# Generalization of Reinforcement Learners with Working and Episodic Memory

**Meire Fortunato**[*]    **Melissa Tan**[*]
**Ryan Faulkner**[*]    **Steven Hansen**[*]    **Adrià Puigdomènech Badia**
**Gavin Buttimore**    **Charlie Deck**    **Joel Z Leibo**    **Charles Blundell**

DeepMind

{meirefortunato, melissatan, rfaulk, stevenhansen,
adriap, buttimore, cdeck, jzl, cblundell}@google.com

([*] Equal Contribution)

## Abstract

Memory is an important aspect of intelligence and plays a role in many deep reinforcement learning models. However, little progress has been made in understanding when specific memory systems help more than others and how well they generalize. The field also has yet to see a prevalent consistent and rigorous approach for evaluating agent performance on holdout data. In this paper, we aim to develop a comprehensive methodology to test different kinds of memory in an agent and assess how well the agent can apply what it learns in training to a holdout set that differs from the training set along dimensions that we suggest are relevant for evaluating memory-specific generalization. To that end, we first construct a diverse set of memory tasks[1] that allow us to evaluate test-time generalization across multiple dimensions. Second, we develop and perform multiple ablations on an agent architecture that combines multiple memory systems, observe its baseline models, and investigate its performance against the task suite.

## 1 Introduction

Humans use memory to reason, imagine, plan, and learn. Memory is a foundational component of intelligence, and enables information from past events and contexts to inform decision-making in the present and future. Recently, agents that utilize memory systems have advanced the state of the art in various research areas including reasoning, planning, program execution and navigation, among others (Graves et al., 2016; Zambaldi et al., 2018; Santoro et al., 2018; Banino et al., 2018; Vaswani et al., 2017; Sukhbaatar et al., 2015).

Memory has many aspects, and having access to different kinds allows intelligent organisms to bring the most relevant past information to bear on different sets of circumstances. In cognitive psychology and neuroscience, two commonly studied types of memory are working and episodic memory. Working memory (Miyake and Shah, 1999) is a short-term temporary store with limited capacity.

In contrast, episodic memory (Tulving and Murray, 1985) is typically a larger autobiographical database of experience (e.g. recalling a meal eaten last month) that lets one store information over a longer time scale and compile sequences of events into episodes (Tulving, 2002). Episodic memory has been shown to help reinforcement learning agents adapt more quickly and thereby boost data

efficiency (Blundell et al., 2016; Pritzel et al., 2017; Hansen et al., 2018). More recently, Ritter et al. (2018) shows how episodic memory can be used to provide agents with context-switching abilities in contextual bandit problems. The transformer (Vaswani et al., 2017) can be viewed as a hybrid of working memory and episodic memory that has been successfully applied to many supervised learning problems.

In this work, we explore adding such memory systems to agents and propose a consistent and rigorous approach for evaluating whether an agent demonstrates generalization-enabling memory capabilities similar to those seen in animals and humans.

One fundamental principle in machine learning is to train on one set of data and test on an unseen holdout set, but it has to date been common in reinforcement learning to evaluate agent performance solely on the training set which is suboptimal for testing generalization (Pineau, 2018). Also, though advances have recently been made on evaluating generalization in reinforcement learning (Cobbe et al., 2018) these have not been specific to memory.

Our approach is to construct a train-holdout split where the holdout set differs from the training set along axes that we propose are relevant specifically to memory, i.e. the scale of the task and precise objects used in the task environments. For instance, if an agent learns in training to travel to an apple placed in a room, altering the room size or apple color as part of a generalization test should ideally not throw it off.

We propose a set of environments that possess such a split and test different aspects of working and episodic memory, to help us better understand when different kinds of memory systems are most helpful and identify memory architectures in agents with memory abilities that cognitive scientists and psychologists have observed in humans.

Alongside these tasks, we develop a benchmark memory-based agent, the Memory Recall Agent (MRA), that brings together previously developed systems thought to mimic working memory and episodic memory. This combination of a controller that models working memory, an external episodic memory, and an architecture that encourages long-term representational credit assignment via an auxiliary unsupervised loss and backpropagation through time that can 'jump' over several time-steps obtains better performance than baselines across the suite. In particular, episodic memory and learning good representations both prove crucial and in some cases stack synergistically.

To summarize, our contribution is to:

- Introduce a suite of tasks that require an agent to utilize fundamental functional properties of memory in order to solve in a way that generalizes to holdout data.
- Develop an agent architecture that explicitly models the operation of memory by integrating components that functionally mimic humans' episodic and working memory.
- Show that different components of our agent's memory have different effectiveness in training and in generalizing to holdout sets.
- Show that none of the models fully generalize outside of the train set on the more challenging tasks, and that the extrapolation incurs a greater level of degradation.

## 2 Task suite overview

We define a suite of 13 tasks designed to test different aspects of memory, with train-test splits that test for generalization across multiple dimensions (https://github.com/deepmind/dm_memorytasks). These include cognitive psychology tasks adapted from PsychLab (Leibo et al., 2018) and DMLab (Beattie et al., 2016), and new tasks built with the Unity 3D game engine (uni) that require the agent to 1) spot the difference between two scenes; 2) remember the location of a goal and navigate to it; or 3) infer an indirect transitive relation between objects. Videos with task descriptions are at https://sites.google.com/view/memory-tasks-suite.

### 2.1 PsychLab

Four tasks in the Memory Tasks Suite use the PsychLab environment (Leibo et al., 2018), which simulates a psychology laboratory in first-person. The agent is presented with a set of one or multiple consecutive images, where each set is called a 'trial'. Each episode has multiple trials.

In **Arbitrary Visuomotor Mapping (AVM)** a series of objects is presented, each with an associated look-direction (e.g. up,left). The agent is rewarded if it looks in the associated direction the next time it sees a given object in the episode (Fig 8(a) in App. B). **Continuous Recognition** presents a series of images with rewards given for correctly indicating whether an image has been previously shown in the episode (Fig 8(b) in App. B). In **Change Detection** the agent sees two consecutive images, separated by a variable-length delay, and has to correctly indicate if the two images differ (Fig 8(c) in App. B). In **What Then Where** the agent is shown a single 'challenge' MNIST digit, then an image of that digit with three other digits, each placed along an edge of the rectangular screen. It next has to correctly indicate the location of the 'challenge' digit (Fig 8(d) in App. B).

## 2.2 3D tasks

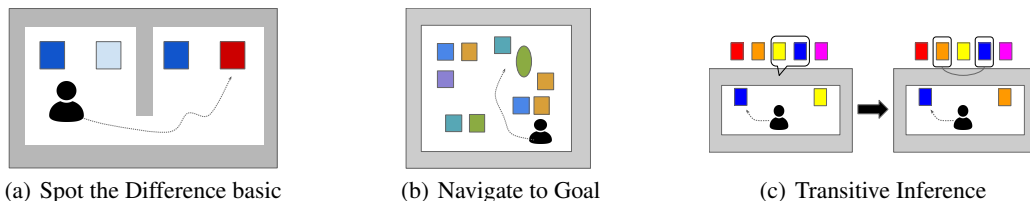

    (a) Spot the Difference basic      (b) Navigate to Goal      (c) Transitive Inference

Figure 1: Task layouts for Spot the Difference, Goal Navigation, and Transitive Inference. In (a), the agent has to identify the difference between the two rooms. In (b), the agent has to go to the goal. which is represented by an oval symbol here and may be visible or not to the agent. In (c), the agent has to go to the higher-valued object in each pair. The value order is given by the transitive chain outside the room. It is shown here solely for illustration; the agent cannot see it.

**Spot the Difference**: This tests whether the agent can correctly identify the difference between two nearly identical scenes (Figure 1(a)). The agent has to move from the first to the second room, with a 'delay' corridor in between. See Fig. 2 for the four different variants.

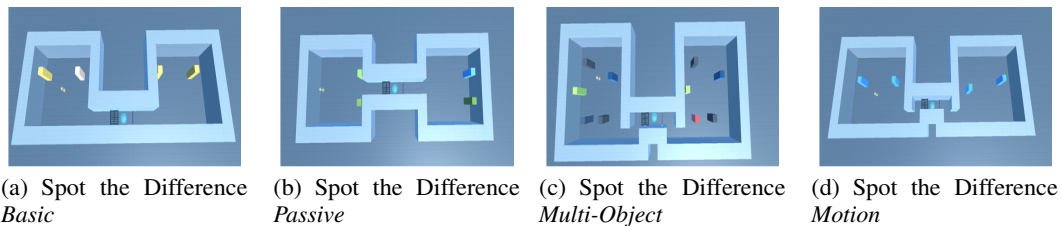

(a) Spot the Difference *Basic*    (b) Spot the Difference *Passive*    (c) Spot the Difference *Multi-Object*    (d) Spot the Difference *Motion*

Figure 2: Spot the Difference tasks. (a) All the tasks in this family are variants of this basic setup, where each room contains two blocks. (b) By placing Room 1's blocks right next to the corridor entrance, we guarantee that the agent will always see them. (c) The number of objects varies. (d) Instead of differing in color between rooms, the altered block follows a different motion pattern.

**Goal Navigation**: This task family was inspired by the Morris Watermaze (Miyake and Shah, 1999) setup used with rodents in behavioral neuroscience. The agent is rewarded every time it successfully reaches the goal; once it gets there it is respawned randomly in the arena and has to find its way back to the goal. The goal location is re-randomized at the start of episode (Fig. 1(b), Fig. 3).

**Transitive Inference**:

This task tests if an agent can learn an overall transitive ordering over a chain of objects, through being presented with ordered pairs of adjacent objects (See Fig. 1(c) and App. B).

## 2.3 Scale and Stimulus Split

To test how well the agent can generalize to holdout data after training, we create per-task holdout levels that differ from the training level along a scale and a stimulus dimension. The scale dimension is intended to capture something about the memory demand of the task: e.g., a task with a longer time delay between events that must be related should be harder than one with a short delay. The stimulus dimension is to guard against trivial overfitting to the particular visual input presented to the input: the memory representation should be more abstract than the particular colour of an object.

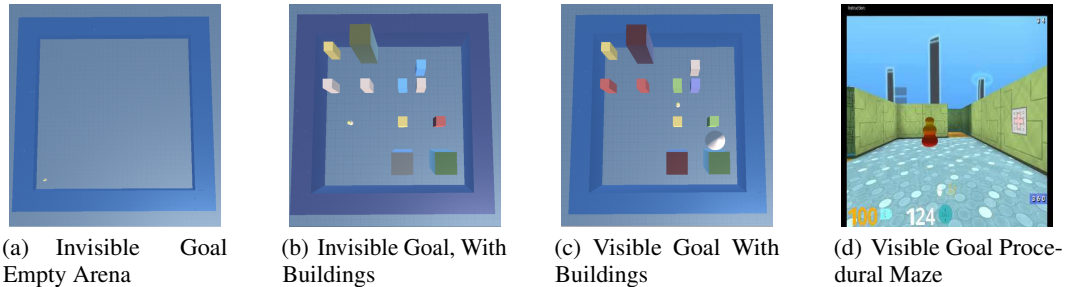

(a) Invisible Goal Empty Arena

(b) Invisible Goal, With Buildings

(c) Visible Goal With Buildings

(d) Visible Goal Procedural Maze

Figure 3: Goal Navigation tasks. (a) The arena has no buildings, agent must navigate by skybox. (b) There are rectangular buildings at fixed, non-randomized locations in the arena. (c) As in (b), but the goal appears as an oval. (d) A visible goal in a procedurally generated maze.

The training level comprises a 'small' and 'large' scale version of the task. When training the agent we uniformly sample between these two scales. As for the holdout levels, one of them – 'holdout-interpolate' – corresponds to an interpolation between those two scales (call it 'medium') and the other, 'holdout-extrapolate', corresponds to an extrapolation beyond the 'large' scale (call it 'extra-large'). Alterations made for each task split and their settings are in Table 2 in App. A.

## 3 The Memory Recall Agent

Our agent, the Memory Recall Agent (MRA), incorporates five components: 1) a pixel-input convolutional, residual network, 2) a working memory, 3) a slot-based episodic memory, 4) an auxiliary contrastive loss for representation learning (van den Oord et al., 2018), 5) a jumpy backpropagation-through-time training regime. Our agent architecture is shown in Figure 4(a). The overall agent is built on top of the IMPALA model (Espeholt et al., 2018) and is trained in the same way with the exceptions described below. Component descriptions are below.

**Pixel Input** Pixel input is fed to a convolutional neural network, as is common in recent agents, followed by a residual block (He et al., 2015). The precise hyper-parameters are given in C.2: we use three convolutional layers followed by two residual layers. The output of this process is $x_t$ in Figure 4(a) and serves as input to three other parts of the network: 1) part of the input to the working memory module, 2) in the formation of keys and queries for the episodic memory, 3) as part of the target for the contrastive predictive coding.

**Working Memory** Working memory is often realized through latent recurrent neural networks (RNNs) with some form of gating, such as LSTMs and Relational Memory architectures (Hochreiter and Schmidhuber, 1997; Santoro et al., 2018). These working memory models calculate the next set of hidden units using the current input and the previous hidden units. Although models which rely on working memory can perform well on a variety of problems, their ability to tackle dependencies and represent variables over long time periods is limited. The short-term nature of working memory is pragmatically, and perhaps unintentionally, reflected in the use of truncated backprop through time and the tendency for gradients through these RNNs to explode or vanish. Our agent uses an LSTM as a model of working memory. As we shall see in experiments, this module is able to perform working memory–like operations on tasks: i.e., learn calculations involving short-term memory. As depicted in Figure 4(a), the LSTM takes as input $x_t$ from the pixel input network and $m_t$ from the episodic memory module. As in Espeholt et al. (2018), the LSTM has two heads as output, producing the policy $\pi$ and the baseline value function $V$. In our architecture these are derived from the output from the LSTM, $h_t$. $h_t$ is also used to form episodic memories, as described below.

**Episodic Memory (MEM)** If our agent only consisted of the working memory and pixel input described above, it would be almost identical to the model in IMPALA (Espeholt et al., 2018), an already powerful RL agent. But MRA also includes a slot-based episodic memory module as that can store values more reliably and longer-term than an LSTM, is less susceptible to the intricacies of gradient propagation, and its fundamental operations afford the agent different abilities (as observed in our experiments). The MEM in MRA has a key-value structure which the agent reads from and writes to at every time-step (see Fig. 4(a)). MRA implements a mechanism to learn how to store summaries

of past experiences and retrieve relevant information when it encounters similar contexts. The reads from memory are used as additional inputs to the neural network (controller), which produces the model predictions. This effectively augments the controller's working memory capabilities with experiences from different time scales retrieved from the MEM, which facilitate learning long-term dependencies, a difficult task when relying entirely on backpropagation in recurrent architectures (Hochreiter and Schmidhuber, 1997; Graves et al., 2016; Vaswani et al., 2017).

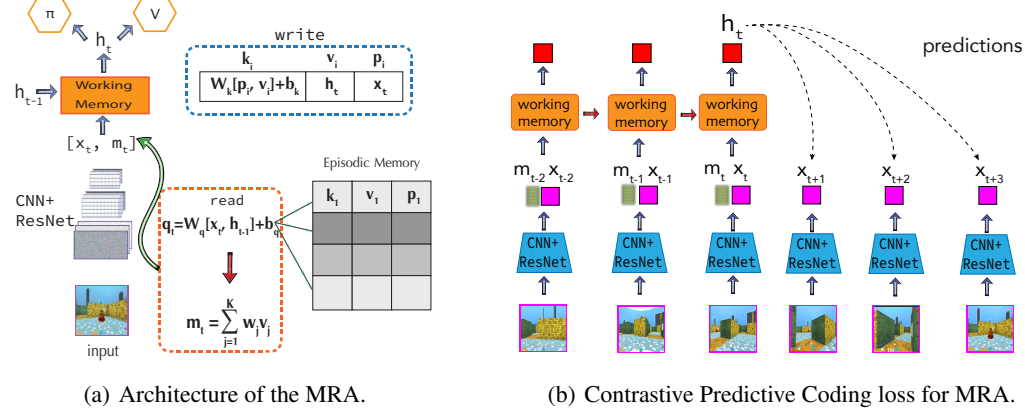

(a) Architecture of the MRA.  (b) Contrastive Predictive Coding loss for MRA.

Figure 4: The Memory Recall Agent (MRA) architecture. Here $p_i$ is the pixel input embedding $x_t$ from step $t$, and $v_i$ is the LSTM hidden state $h_t$. $k_i$ is the key used for reading; we compute it from $p_i$ and $v_i$. $q_t$ is the query we use to compare against keys to find nearest neighbors.

The MEM has a number of slots, indexed by $i$. Each slot stores activations from the pixel input network and LSTM from previous times $t_i$ in the past. The MEM acts as a fixed-size circular (first-in-first-out) buffer: New keys and values are added, overwriting the least recently added entry if there are no unused slots available. The contents of the episodic memory buffer is wiped at the end of each episode.

**Memory Writing**  Crucially, writing to episodic memory is done without gradients. At each step a free slot is chosen for writing, denoted $i$. Next, the following is stored:

$$p_i \leftarrow x_t; \quad v_i \leftarrow h_t; \quad k_i \leftarrow W_k[p_i, v_i] + b_k \tag{1}$$

where $p_i$ is the pixel input embedding from step $t$ and $v_i$ is the LSTM hidden state (if the working memory is something else, e.g. a feedforward, this would be the output activations). $k_i$ is the key, used for reading (described below), computed as a simple linear function of the other two values stored. Caching the key speeds up memory reads significantly. However, the key can become stale as the weights and biases, $W_k$ and $b_k$ are learnt (the procedure for learning them is described below under Jumpy Backpropagation). In our experiments we did not see an adverse effect of this staleness.

**Memory Reading**  The agent uses a form of dot-product attention (Bahdanau et al., 2015) over its MEM, to select the most relevant events to provide as input $m_t$ to the LSTM. The query $q_t$ is a linear transform of the pixel input embedding $x_t$ and the LSTM hidden state from the previous time-step $h_{t-1}$, with weight $W_q$ and bias $b_q$.

$$q_t = W_q[x_t, h_{t-1}] + b_q \tag{2}$$

The query $q_t$ is then compared against the keys in MEM as in Pritzel et al. (2017): Let $(p_j, v_j, k_j)$, $1 \leq j \leq K$ be the $K$ nearest neighbors to $q_t$ from MEM, under an L2 norm between $k_j$ and $q_t$.

$$m_t = \sum_{j=1}^{K} w_j v_j \qquad where \quad w_j \propto \frac{1}{\epsilon + ||q_t - W_k[p_j, v_j] - b_k||_2^2} \tag{3}$$

We compute a weighted aggregate of the values ($v_j$) of the $K$ nearest neighbors, weighted by the inverse of each neighbor-key's distance to the query. Note that the distance is re-calculated from values stored in the MEM, via the linear projection $W_k, b_k$ in (1). We concatenate the resulting weighted aggregate memory $m_t$ with the embedded pixel input $x_t$, and pass it as input to the working memory as shown in Figure 4(a).

**Jumpy backpropagation**   We now turn to how gradients flow into memory writes. Full backprop-agation can become computationally infeasible as this would require backpropagation into every write that is read from and so on. Thus as a new $(p_i, v_i, k_i)$-triplet is added to the MEM, there are trade-offs to be made regarding computational complexity versus performance of the agent. To make it more computationally tractable, we place a stop-gradient in the memory write. In particular, the write operation for the key in (1) becomes:

$$k_i \leftarrow W_k[\text{SG}(p_i), \text{SG}(v_i)] + b_k \tag{4}$$

where $\text{SG}(\cdot)$ denote that the gradients are stopped. This allows the parameters $W_k$ and $b_k$ to receive gradients from the loss during writing and reading, while at the same time bounding the computational complexity as the gradients do not flow back into the recurrent working memory (or via that back into the MEM). To re-calculate the distances, we want to use these learnt parameters rather than, say, random projection, so we need to store the arguments $x_t$ and $h_t$ of the key-generating linear transform $W_k, b_k$ for all previous time-steps. Thus in the MEM we store the full $(p_i, v_i, k_i)$-triplet, where $p_i = x_{t_i}$, $v_i = h_{t_i}$ and $t_i$ is the step that write $i$ was made. We call this technique 'jumpy backpropagation' because the intermediate steps between the current time-step $t$ and the memory write step $t_i$ are not taken into account in the gradient updates.

This approach is similar to Sparse Attentive Backtracking (Ke et al., 2018, SAB) which uses sparse replay by passing gradients only through memories selected as relevant at each step. Our model differs in that it does not have a fixed chunking scheme and does not do full backpropagation through the architecture (which in our case becomes quickly intractable). Our approach has minimal computational overhead as we only recompute the keys for the nearest neighbors.

**Auxiliary Unsupervised Losses**   An agent with good memory provides a good basis for forming a rich representation of the environment, as it captures a history of the states visited by the agent. This is the primary basis for many rich probabilistic state representations in reinforcement learning such as belief states and predictive state representations (Littman and Sutton, 2002). Auxiliary unsupervised losses can significantly improve agent performance (Jaderberg et al., 2016). Recently it has been shown that agents augmented with one-step contrastive predictive coding (van den Oord et al., 2018, CPC) can learn belief state representations of the environment (Guo et al., 2018). Thus in MRA we combine the working and episodic memory mechanisms listed above with a CPC unsupervised loss to imbue the agent with a rich state representation. The CPC auxiliary loss is added to the usual RL losses, and is of the following form:

$$\sum_{\tau=1}^{N} \text{CPCLoss}\left[h_t; x_{t+1}, x_{t+2}, \ldots, x_{t+\tau}\right] \tag{5}$$

where CPCLoss is from van den Oord et al. (2018), $h_t$ is the working memory hidden state, and $x_{t+\tau}$ is the encoding pixel input at $\tau$ steps in the future. $N$ is the number of CPC steps (typically 10 or 50 in our experiments). See Figure 4(b) for an illustration and further details and equations elaborating on this loss in App. C.3.

Reconstruction losses have also been used as an auxiliary task (Jaderberg et al., 2016; Wayne et al., 2018) and we include this as a baseline in our experiments. Our reconstruction baseline minimizes the L2 distance between the predicted reward and predicted pixel input and the true reward and pixel input, using the working memory state $h_t$ as input. Details of this baseline are given in App. C.4.

## 4   Experiments

**Setup**   We ran 10 ablations on the MRA architecture, on the training and the two holdout levels:

- Working Memory component: Either feedforward neural network ('FF' for short) or LSTM. The LSTM-only baseline corresponds to IMPALA (Espeholt et al., 2018).
- With or without using episodic memory module ('MEM').
- With or without auxiliary unsupervised loss (either CPC or reconstruction loss ('REC')).
- With or without jumpy backpropagation, for MRA (i.e. LSTM + MEM + CPC)

Given that the experiments are computationally demanding, we only performed small variations within as part of our hyper-parameter tuning process for each task (see App. D).

We hypothesize that in general the agent should perform the best in training, somewhat worse on the holdout-interpolation level and the worst on the holdout-extrapolation level. That is, we expect to see a *generalization gap*. Our results validated this hypothesis for the tasks that were much harder for agents than for humans.

## 4.1 Full comparison

We computed human-normalized scores (details in App. B) and plotted them into a heatmap (Fig 5) sorted such that the model with the highest train scores on average is the top row and the task with highest train scores on average is the leftmost column. The heatmap suggests that the MRA architecture, LSTM + MEM + CPC, broadly outperforms the other models (App. B Table 3). This ranking was almost always maintained across train and holdout levels, despite MRA performing worse than the LSTM-only baseline on *What Then Where*. *What Then Where* was one of the tasks where all models did poorly, along with *Spot the Difference: Multi-Object*, *Spot the Difference: Multi-Object*, *Spot the Difference: Multi-Object* (rightmost columns in heatmap). At the other end of the difficulty spectrum, LSTM + MEM had superhuman scores on *Visible Goal Procedural Maze* in training and on *Transitive Inference* in training and holdout, and further adding CPC or REC boosted the scores even higher.

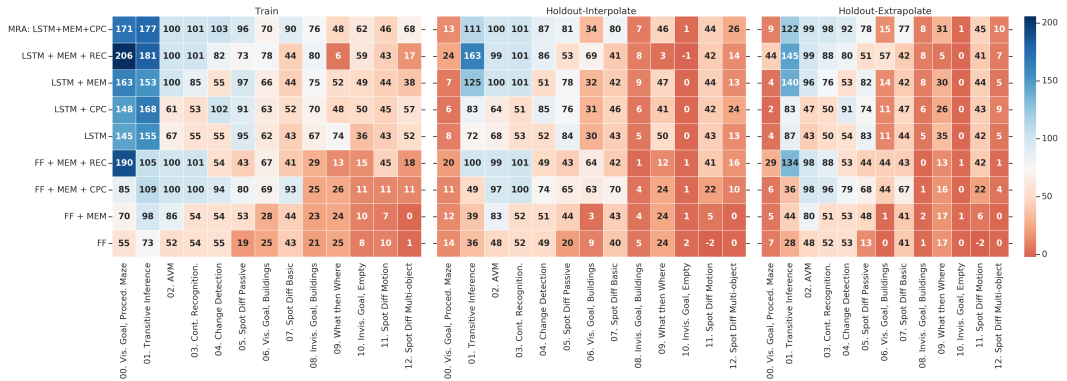

Figure 5: Heatmap of ablations per task sorted by normalized score for Train, Holdout-Interpolate, Holdout-Extrapolate. The same plot with standard errors is in App. B Fig. 14.

## 4.2 Results

Different memory systems worked best for different kinds of tasks, but the MRA architecture's combination of LSTM + MEM + CPC did the best overall on training and holdout (Fig. 6). Removing jumpy backpropagation from MRA hurt performance in five Memory Suite tasks (App. B Fig. 10), while performance was the same in the remaining ones (App. B Fig. 11 and 12).

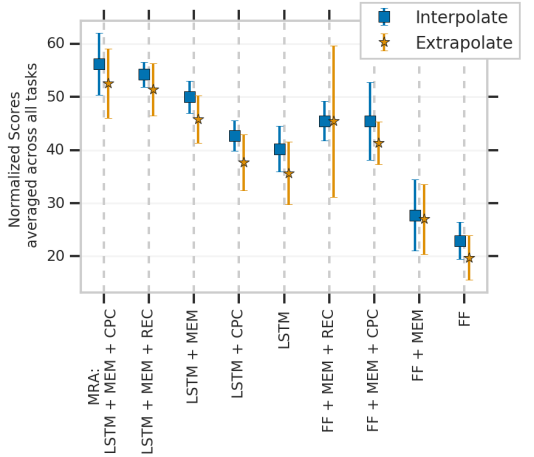

Figure 6: Normalized scores averaged across tasks.

**Generalization gap widens as task difficulty increases** The hypothesized generalization gap was minimal for some tasks e.g. *AVM* and *Continuous Recognition* but significant for others e.g. *What Then Where* and *Spot the Difference: Multi-Object* (Fig 7). We observed that the gap tended to be wider as the task difficulty went up, and that in PsychLab, the two tasks where the scale was the number of trials seemed to be easier than the other two tasks where the scale was the delay duration.

**MEM critical on some tasks, is enhanced by auxiliary unsupervised loss** Adding MEM improved scores on nine tasks in training, six in holdout-interpolate, and six in holdout-extrapolate. Adding MEM alone, without an auxiliary unsupervised loss, was enough to improve scores on *AVM*

and *Continuous Recognition*, all Spot the Difference tasks except *Spot the Difference: Multi-Object*, all Goal Navigation tasks except *Visible Goal Procedural Maze*, and also for *Transitive Inference*.

Adding MEM helped to significantly boost holdout performance for *Transitive Inference*, *AVM*, and *Continuous Recognition*. For the two PsychLab tasks this finding was in line with our expectations, since they both can be solved by memorizing single images and determining exact matches and thus an external episodic memory would be the most useful. For *Transitive Inference*, in training MEM helped when the working memory was FF but made little difference on an LSTM, but on holdout MEM helped noticeably for both FF and LSTM. In *Change Detection* and *Multi-Object*, adding MEM alone had little or no effect but combining it with CPC or REC provided a noticeable boost.

**Synergistic effect of MEM + CPC, for LSTM**    On average, adding either the MEM + CPC stack or MEM + REC stack to any working memory appeared to improve the agent's ability to generalize to holdout levels (Fig. 6). Interestingly, on several tasks we found that combining MEM + CPC had a synergistic effect when the working memory was LSTM: The performance boost from adding MEM + CPC was larger than the sum of the boost from adding MEM or CPC alone. We observed this phenomenon in seven tasks in training, six in holdout-interpolate, and six in holdout-extrapolate. Among these, the tasks where there was MEM + CPC synergy across training, holdout-interpolate, and holdout-extrapolate were: the easiest task, *Visible Goal Procedural Maze*; *Visible Goal with Buildings*; *Spot the Difference: Basic*; and the hardest task, *Spot the Difference: Multi-Object*.

**CPC vs. REC**    CPC was better than REC on all Spot the Difference tasks, and the two harder PsychLab tasks *Change Detection* and *What Then Where*. On the other two PsychLab tasks there was no difference between CPC and REC. However, REC was better on all Goal Navigation tasks except *Invisible Goal Empty Arena*. When averaged out, REC was more useful when the working memory was FF, but CPC was more useful for an LSTM working memory.

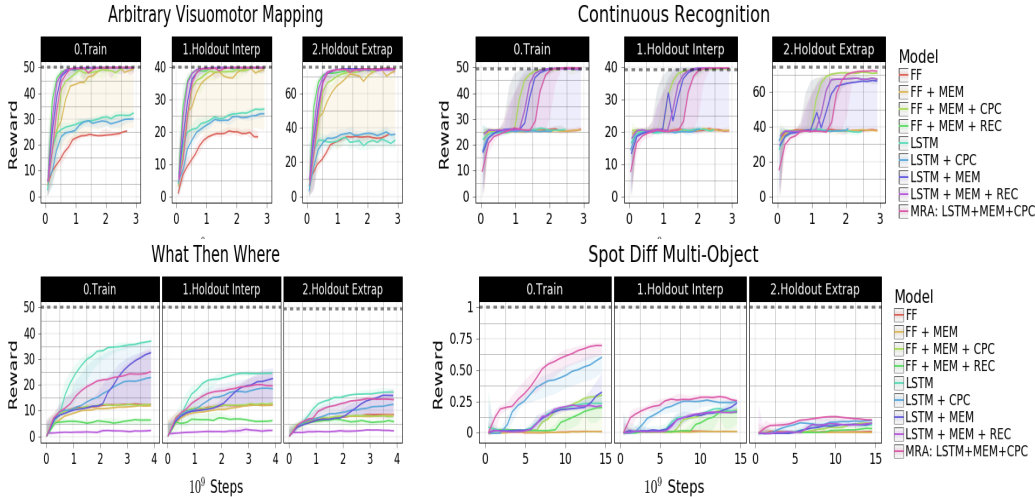

Figure 7: Generalization gap is smaller for *AVM* and *Continuous Recognition*, larger for *What Then Where* and *Spot the Difference: Multi-Object*. Dotted lines indicate human baseline scores. See other curves in App. B Fig. 13.

## 5   Discussion & Future Work

We constructed a diverse set of environments [2] to test memory-specific generalization, based on tasks designed to identify working memory and episodic memory in humans, and also developed an agent that demonstrates many of these cognitive abilities. We propose both a testbed and benchmark for further work on agents with memory, and demonstrate how better understanding the memory and generalization abilities of reinforcement learning agents can point to new avenues of research to improve agent performance and data efficiency. There is still room for improvement on the trickiest tasks in the suite where the agent fared relatively poorly. In particular, solving *Spot the Difference:*

*Motion* might need a generative model that enables forward planning to imagine how future motion unrolls (e.g., (Racanière et al., 2017)). Our results indicate that adding an auxiliary loss such as CPC or reconstruction loss to an architecture that already has an external episodic memory improves generalization performance on holdout sets, sometimes synergistically. This suggests that existing agents that use episodic memory, such as DNC and NEC, could potentially boost performance by implementing an additional auxiliary unsupervised loss.

## Acknowledgements

We would like to thank Jessica Hamrick, Jean-Baptiste Lespiau, Frederic Besse, Josh Abramson, Oriol Vinyals, Federico Carnevale, Charlie Beattie, Piotr Trochim, Piermaria Mendolicchio, Aaron van den Oord, Chloe Hillier, Tom Ward, Ricardo Barreira, Matthew Mauger, Thomas Köppe, Pauline Coquinot and many others at DeepMind for insightful discussions, comments and feedback on this work.

## Footnotes

[1]https://github.com/deepmind/dm_memorytasks. Videos available at https://sites.google.com/view/memory-tasks-suite

[2]Available at https://github.com/deepmind/dm_memorytasks.

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
