[Supplementary Material]

# A    Level descriptions and further experimental findings

As described in Section 2.3, for each task in the Suite we construct a small training level, a large training level, a 'holdout-interpolation' level and a 'holdout-extrapolation' level.

During training the environment uniformly samples from the small and large training levels. The interpolation level has a scale somewhere in between 'small' and 'large' while the extrapolation level corresponds to 'extra-large' (Table 1). A summary of the alterations made for each task split is in Table 2. The settings used in each level per task are described below.

Table 1: **Overall structure for scale and stimulus split.**

| Scale \ Stimuli | Training set | Holdout set |
|---|---|---|
| Small | Used for training | — |
| Medium | — | Used for interpolation |
| Large | Used for training | — |
| Extra-large | — | Used for extrapolation |

The dashed ('—') settings in Table 1 are not reported nor used, since they lack a clear interpretation in terms of generalization.

Table 2: **Scale and stimulus alterations across task families**

| Task | Scale | Stimulus |
|---|---|---|
| AVM | Number of trials | Image |
| Continuous Recognition | Number of trials | Image |
| Change Detection | Delay study/test | Color |
| What Then Where | Delay study/query | Digit image |
| Spot Diff Basic | Corridor delay | Color |
| Spot Diff Passive | Corridor delay duration | Color |
| Spot Diff Multi-object | Number of objects | Color |
| Spot Diff Motion | Corridor delay | Motion pattern |
| All Goal Navigation tasks | Arena size | Goal spawn |
| Transitive Inference | Length of transitive chain | Object color |

## A.1    PsychLab

Our Memory Tasks Suite has four PsychLab tasks: Arbitrary Visuomotor Mapping (AVM), Continuous Recognition, Change Detection and What Then Where. The description of each task is found in Figure 8. Videos with agent play and the https://sites.google.com/view/memory-tasks-suite.

**Scale**    Either number of trials per episode or delay duration.

For *Arbitrary Visuomotor Mapping* and *Continuous Recognition*, every episode lasts at most 300 seconds, except for the Extrapolate level where the cap is set to 450 seconds to accommodate the larger number of trials. In *Change Detection* an episode lasts at most 300 seconds, while for *What Then Where* it is 600 seconds.

| Scale \ Task | AVM and Cont. Recog.: Trials per episode | Change Detection: delay (seconds) | What Then Where: delay (seconds) |
|---|---|---|---|
| Small | 50 | 2, 4, 8 | 4, 8 |
| Interpolate | 40 | 16, 32 | 16, 64 |
| Large | 50 | 64, 128 | 32, 128 |
| Extrapolate | 75 | 130, 150, 200, 250 | 132, 156, 200, 256 |

**Stimulus**    Either color set or image set.

(a) Arbitrary Visuomotor Mapping (AVM)

(b) Continuous Recognition

(c) Change Detection

(d) What Then Where

Figure 8: All PsychLab tasks have multiple trials within an episode. Each trial consists of a single image being displayed on the panel. In (a), when the agent sees an image for the first time, the associated direction is indicated on the screen (green box on the left). By executing the indicated pattern, the agent receives a reward. When the agent is presented with an image it has already seen during the episode, the associated direction is no longer indicated (middle), and the agent must remember it from its previous experience in order to get a reward (right). In (b), the agent is shown a pattern (left), and after a delay (middle), a second pattern is shown (right). The agent has to indicate if there was a change between the two patterns or not by looking right or left, respectively. The delay period separating the two patterns varies in length. In (c), the agent indicates if it has seen the image in the current episode by looking left or right, respectively. In (d), in the 'what' study phase, an MNIST digit is displayed (left). In the 'where' study period, four distinct MNIST digits are displayed including the one from the 'what' period (middle). In the test phase (right), the agent must remember what digit was displayed in the 'what' period, see where it is located during the study where period, and then respond by looking to that location. In this example it has to look left.

| Task | AVM and Cont. Recog | Change Detection | What Then Where |
|---|---|---|---|
| Stimulus | Different images | Color set | MNIST digits |
| Training | Images with even ID | Amethyst, Caramel, Honeydew, Jade, Mallow | 0, 1, 2, 3, 4 |
| Holdout | Images with odd ID | Yellow, Lime, Pink, Sky, Violet | 5, 6, 7, 8, 9 |

### A.1.1 PsychLab: main experimental findings

**AVM:** in this task, the agent must remember associations between images and specific movement patterns (Figure 8 (a)).

The most useful component turned out to be MEM. This is in line with earlier findings that an external episodic memory is a prerequisite for solving AVM (Wayne et al., 2018). Adding an auxiliary loss helped when the controller was FF but made no difference for an LSTM. Also, choosing between CPC or REC for auxiliary unsupervised loss did not make a major difference for either controller.

**Continuous Recognition:** in this task, the agent must remember if it has seen a particular image before by looking left or right (Figure 8 (b)).

MEM was the most useful component when added to an LSTM, but made no difference when added alone to an FF controller. However, adding a stack of MEM plus either CPC or REC provided a substantial performance boost for both FF and LSTM.

**Change Detection:**  in this task, agent sees two images separated by a delay and has to correctly indicate if the two images are different (Figure 8 (c)).

CPC brought the largest benefit. Interestingly the addition of MEM to the FF baseline actually hurt performance slightly, and made no difference for LSTM.

**What Then Where:**  this task consists of a 'what' and 'where' study phase, followed by a test phase where the agent must remember what image was displayed and where it was located (Figure 8 (d)).

This was the trickiest task in the Psychlab family. This task was an outlier in the sense that unlike any other task in the suite, the LSTM baseline beat all other models. The worst additional component was REC which dragged down performance to below random.

## A.2  Spot the Difference (SD)

The tasks were built in Unity, and each episode lasts 120 seconds except for *Spot the Difference: Motion* which has a 240-second timeout.

**Scale**  Either corridor delay duration or number of objects in room.

In Spot the Difference Multi-Object, Room 2 has the exact same number of objects as Room 1.

| Scale \ Task | SD Basic, Passive and Motion: Corridor delay (seconds) | SD Multi-Object: Number of objects in Room 1 |
|---|---|---|
| Small | 0 | 2 or 3 |
| Interpolate | 5 | 4 |
| Large | 10 | 5 or 6 |
| Extrapolate | 15 | 7 |

**Stimulus**  Either color set or motion pattern set.

| Task | SD Basic, Passive and Motion | SD Multi-Object |
|---|---|---|
| Stimulus | Color Set | Motion Pattern Set |
| Training | Red, Green, Blue, White, Slate | Circle, Square, Five-point star, Hexagon Linear along X-axis, Linear along Y = X diagonal |
| Holdout | Yellow, Brown, Pink, Orange, Purple | No motion, Triangle, Pentagon, Figure-eight Linear along Y-axis, Linear along Y = -X diagonal |

### A.2.1  Spot the Diff: main experimental findings

Every task in this family consists of two rooms connected by a short corridor. There is a set of gates in the middle of the corridor that can trap the agent there for a configurable delay duration.

**Basic**  In the basic Spot the Difference task, where the agent is not forced to see any of the blocks in Room 1 before it goes to the next room, adding MEM alone to the controller had minimal effect, and using REC with MEM also did not make much difference. Adding CPC to an LSTM helped performance but it turned out that using the combination of MEM + CPC provided the biggest gain and was synergistic.

**Passive**  In this task the agent is guaranteed to see the two blocks in the first room before it enters the second room. Adding MEM alone to the controller made the biggest positive difference, which makes sense since that hypothetically would make it possible for the agent to solve the task by remembering a single snapshot. CPC helped when added to FF together with MEM, but hurt when added to LSTM alone. REC helped performance when added to FF + MEM, but not as much as CPC did in that case, and actually hurt performance when added to LSTM + MEM.

**Motion**  Nothing did well on train or holdout sets, and curves took longer to take off in general. This is likely due to the highly challenging nature of the task, which requires the agent to memorize 3D motion patterns traced out over some time period by multiple objects and then compare motion patterns against each other. Results would potentially be improved by hyperparameter tuning or further improvements to agent architecture.

**Multi-object**   This was the hardest task in the family, and nothing did well here either. This could be due to there being a variable number of objects in each room, rather than always exactly two objects per room. When added by itself to a controller MEM either had no effect or hurt performance. The combined synergistic stack of MEM + CPC was the most useful addition on this task when the working memory was LSTM. That said, no models fared well on Holdout-Interpolate and Holdout-Extrapolate for this task.

## A.3   Navigate to Goal

These tasks are in Unity and have an episode timeout of 200 seconds, except *Visible Goal Procedural Maze* which is a modification of DMLab's *Explore Goal Locations* task and has episodes lasting 120 seconds each.

**Scale**   Size of square arena, in terms of in-game metric units.

| Scale \ Task | Visible Goal Procedural Maze: Arena Size | All but Visible Goal Procedural Maze: Arena Size |
|---|---|---|
| Small | $11 \times 11$ | $10 \times 10$ |
| Interpolate | $15 \times 15$ | $15 \times 15$ |
| Large | $21 \times 21$ | $20 \times 20$ |
| Extrapolate | $27 \times 27$ | $25 \times 25$ |

**Stimulus**   Goal spawn region.

| Stimulus \ Task | Visible Goal Procedural Maze: Goal spawn region | All but Visible Goal Procedural Maze: Goal spawn region |
|---|---|---|
| Training | North half | Northwest and southeast quadrants |
| Holdout | South half | The other two quadrants |

### A.3.1   Navigate to Goal: main experimental findings

Using an auxiliary unsupervised reconstruction loss to learn high-quality representations turned out to be the most useful component for this task family.

We also observed that in successful models such as LSTM + MEM + CPC, which is the MRA architecture, the agent is able to do better than simply memorizing a route to the invisible goal. Rather, it learns the location of the goal, and the time it takes to reach the goal location grows shorter every time it respawns within an episode (see example trajectory in Fig 9(a) and time-to-goal plot in Fig 9(b)).

**Visible Goal Procedural Maze**   Using REC with LSTM + MEM performed the best here, and FF + MEM + REC was the next best. The MEM + CPC stack was a distant runner-up compared with the MEM + REC stack for both controllers.

**Visible Goal With Buildings**   Like in the other Visible Goal task, LSTM + MEM + REC was the most successful model. MEM was slightly more helpful than CPC when used in conjunction with an LSTM (we did not have bandwidth to run the FF + CPC ablation). MEM + CPC also had a synergistic effect when stacked with an LSTM.

**Invisible Goal With Buildings**   Adding MEM + REC was the most useful, for both FF and LSTM.

**Invisible Goal Empty Arena**   This task can be expected to be the most difficult in the family due to the relative sparsity of visual spatial cues. Adding MEM alone to a controller always helped slightly. REC helped more than CPC did when used with an FF controller but for an LSTM controller CPC had a slight edge.

(a) Routes taken by MRA agent in one episode

(b) Timesteps taken to reach goal

Figure 9: Trajectories and time-to-goal for *Invisible Goal with Buildings*. In (a), our MRA (LSTM + MEM + CPC) agent learns to take increasingly shorter routes to the goal. Note: The end-points of each trial trajectory appear to be in slightly different locations. This is because the goal is on a map tile rather than a single coordinate, and also due to manual adjustments we made to account for the agent avatar in Unity continuing to move for a small number of frames immediately after reaching the goal but before it is respawned. In (b), the number of time-steps taken per trial is plotted for Train, Holdout-Interpolate, Holdout-Extrapolate, along with standard error bars. Note: Some points at the rightmost end of each curve will have no error bar if there was only one data point.

## A.4 Transitive Inference

The task was built in Unity and has an episode timeout of 200 seconds.

**Scale:** Number of objects in transitive chain. **Stimulus:** Color set.

| Scale | Transitive chain length | | Stimulus | Color set |
|---|---|---|---|---|
| Small | 5 | | Training | Red, Green, Blue, White, Black, |
| Interpolate | 6 | | | Pink, Orange, Purple, Grey, Tan |
| Large | 7 | | Holdout | Slate, Yellow, Brown, Lime, Magenta |
| Extrapolate | 8 | | | Mint, Navy, Olive, Teal, Turquoise |

### A.4.1 Transitive Inference: main experimental findings

Transitive inference is a form of reasoning where one infers a relation between items that have not been explicitly directly compared to each other. In humans, performance on probe pairs and anchor pairs with symbolic distance of greater than one excluding anchor objects tends to correlate with awareness of the implied hierarchy (Smith and Squire, 2005).

As an illustrative example: Given a 'transitive chain' of five objects *A, B, C, D, E* where we assume *A* is the lowest-valued object and *E* the highest, we begin with a demonstration phase in which we present the agent with pairs of adjacent objects *<A, B>, <B, C>, <C, D>, <D, E>* .

In this demo phase we scramble the order in which the pairs are presented and also scramble the objects in the pair such that an agent may see *<D, C>* followed by *<A, B>*, etc. The pairs are presented one at a time, and the agent needs to correctly identify the higher-valued object in the current pair in order to proceed to seeing the next pair.

Once the demo phase is completed, we show the agent a single, possibly-scrambled challenge pair. This challenge pair always consists of the object second from the left and the object second from the right in the transitive chain, in this case *<B, D>*. The agent's task is again to go to the higher-valued object.

In our results, we found that stacking MEM with auxiliary loss was crucial. For an FF controller CPC was more useful than REC, but for LSTM it was the other way round. Also, although both LSTM + MEM + CPC and LSTM + MEM + REC achieved normalized scores that were not too far apart, REC was more data-efficient and took off earlier than the former. We observed a synergistic effect when combining MEM with CPC for an LSTM, but that was still outdone by using MEM + REC.

## A.5 Jumpy Backpropagation (JB) ablation

We studied the impact of having Jumpy Backpropagation (JB) as described in Section 3. In Fig 10, we can see the set of tasks where adding the JB yields improvements on performance both at training time and on the holdout test levels. Figures 11 and 12 show the performance on the remaining levels from the Memory Task Suite, where having the JB feature did not hurt performance. We conclude that JB is an important component of the MRA architecture.

Figure 10: Comparison between MRA (LSTM + MEM +CPC) and its version without the jumpy backpropagation feature on MEM: LSTM + MEM (no JB) + CPC. Here we show the tasks where JB yields improvements on performance both at training time and on the holdout test levels. The dotted lines indicate human baseline scores for each task.

## Continuous Recognition

## Spot Diff Passive

## Spot Diff Multi-Object

## Spot Diff Motion

## Invisible Goal Empty Arena

Figure 11: [1/2] Comparison between MRA and its version without the jumpy backpropagation (JB) feature. Here we show the tasks where JB makes little difference on performance. The dotted lines indicate human baseline scores for each task.

Figure 12: [2/2] Comparison between MRA and its version without the jumpy backpropagation (JB) feature. Here we show the tasks where JB makes little difference on performance. The dotted lines indicate human baseline scores for each task.

## A.6 Agent Performance Curves

In this session we show training and test curves for all models in all tasks. The dotted lines indicate human baseline scores for each task.

# Arbitrary Visuomotor Mapping

# Continuous Recognition

# Change Detection

# What Then Where

# Spot Diff Passive

## Spot Diff Basic

## Spot Diff Multi-Object

## Spot Diff Motion

## Invisible Goal Empty Arena

## Invisible Goal With Buildings

Figure 13: Training and test curves for all models in all tasks. Dotted lines indicate human baseline scores for each task.

**Train**

| Model | 00. Vis. Goal, Proced. Maze | 01. Transitive Inference | 02. AVM | 03. Cont. Recognition. | 04. Change Detection | 05. Spot Diff Passive | 06. Vis. Goal, Buildings | 07. Spot Diff Basic | 08. Invis. Goal, Buildings | 09. What then Where | 10. Invis. Goal, Empty | 11. Spot Diff Motion | 12. Spot Diff Multi-object |
|---|---|---|---|---|---|---|---|---|---|---|---|---|---|
| MRA: LSTM+MEM+CPC | 171 ± 1 | 177 ± 3 | 100 ± 0 | 101 ± 0 | 103 ± 1 | 96 ± 1 | 70 ± 1 | 90 ± 7 | 76 ± 1 | 48 ± 11 | 62 ± 3 | 46 ± 1 | 68 ± 2 |
| LSTM + MEM + REC | 206 ± 3 | 181 ± 0 | 100 ± 0 | 101 ± 0 | 82 ± 12 | 73 ± 13 | 78 ± 1 | 44 ± 1 | 80 ± 0 | 6 ± 0 | 59 ± 0 | 43 ± 2 | 17 ± 4 |
| LSTM + MEM | 163 ± 3 | 153 ± 7 | 100 ± 0 | 85 ± 14 | 55 ± 0 | 97 ± 1 | 66 ± 1 | 44 ± 0 | 75 ± 0 | 52 ± 12 | 49 ± 1 | 44 ± 1 | 38 ± 5 |
| LSTM + CPC | 148 ± 1 | 168 ± 10 | 61 ± 0 | 53 ± 1 | 102 ± 1 | 91 ± 1 | 63 ± 1 | 52 ± 3 | 70 ± 2 | 48 ± 12 | 50 ± 2 | 45 ± 0 | 57 ± 3 |
| LSTM | 145 ± 4 | 155 ± 14 | 67 ± 0 | 55 ± 1 | 55 ± 0 | 95 ± 0 | 62 ± 1 | 43 ± 1 | 67 ± 1 | 74 ± 0 | 36 ± 1 | 43 ± 0 | 52 ± 20 |
| FF + MEM + REC | 190 ± 10 | 105 ± 33 | 100 ± 0 | 101 ± 0 | 54 ± 0 | 43 ± 0 | 67 ± 2 | 41 ± 1 | 29 ± 1 | 13 ± 1 | 15 ± 1 | 45 ± 1 | 18 ± 2 |
| FF + MEM + CPC | 85 ± 2 | 109 ± 17 | 100 ± 0 | 100 ± 0 | 94 ± 0 | 80 ± 1 | 69 ± 1 | 93 ± 3 | 25 ± 1 | 26 ± 1 | 11 ± 0 | 11 ± 9 | 11 ± 8 |
| FF + MEM | 70 ± 4 | 98 ± 15 | 86 ± 12 | 54 ± 0 | 54 ± 0 | 53 ± 5 | 28 ± 3 | 44 ± 1 | 23 ± 2 | 24 ± 0 | 10 ± 0 | 7 ± 5 | 0 ± 0 |
| FF | 55 ± 1 | 73 ± 6 | 52 ± 1 | 54 ± 1 | 55 ± 0 | 19 ± 9 | 25 ± 2 | 43 ± 0 | 21 ± 1 | 25 ± 0 | 8 ± 1 | 10 ± 4 | 1 ± 1 |

**Holdout-Interpolate**

| Model | 00. Vis. Goal, Proced. Maze | 01. Transitive Inference | 02. AVM | 03. Cont. Recognition. | 04. Change Detection | 05. Spot Diff Passive | 06. Vis. Goal, Buildings | 07. Spot Diff Basic | 08. Invis. Goal, Buildings | 09. What then Where | 10. Invis. Goal, Empty | 11. Spot Diff Motion | 12. Spot Diff Multi-object |
|---|---|---|---|---|---|---|---|---|---|---|---|---|---|
| MRA: LSTM+MEM+CPC | 13 ± 0 | 111 ± 20 | 100 ± 0 | 101 ± 0 | 87 ± 0 | 81 ± 1 | 34 ± 0 | 80 ± 0 | 7 ± 0 | 46 ± 4 | 1 ± 0 | 44 ± 0 | 26 ± 0 |
| LSTM + MEM + REC | 24 ± 1 | 163 ± 2 | 99 ± 1 | 101 ± 0 | 86 ± 4 | 53 ± 6 | 69 ± 0 | 41 ± 1 | 8 ± 1 | 3 ± 1 | -1 ± 0 | 42 ± 1 | 14 ± 2 |
| LSTM + MEM | 7 ± 1 | 125 ± 5 | 100 ± 0 | 101 ± 0 | 51 ± 1 | 78 ± 4 | 32 ± 1 | 42 ± 1 | 9 ± 0 | 47 ± 5 | 0 ± 0 | 44 ± 0 | 13 ± 7 |
| LSTM + CPC | 6 ± 0 | 83 ± 6 | 64 ± 1 | 51 ± 1 | 85 ± 0 | 76 ± 0 | 31 ± 0 | 46 ± 3 | 6 ± 0 | 41 ± 8 | 0 ± 0 | 42 ± 0 | 24 ± 1 |
| LSTM | 8 ± 1 | 72 ± 14 | 68 ± 1 | 53 ± 0 | 52 ± 1 | 84 ± 1 | 30 ± 0 | 43 ± 1 | 5 ± 1 | 50 ± 1 | 0 ± 0 | 43 ± 1 | 13 ± 7 |
| FF + MEM + REC | 20 ± 3 | 100 ± 13 | 99 ± 0 | 101 ± 0 | 49 ± 1 | 43 ± 0 | 64 ± 1 | 42 ± 2 | 1 ± 0 | 12 ± 1 | 1 ± 0 | 41 ± 1 | 16 ± 1 |
| FF + MEM + CPC | 11 ± 1 | 49 ± 23 | 97 ± 1 | 100 ± 0 | 74 ± 1 | 65 ± 3 | 63 ± 2 | 70 ± 4 | 4 ± 0 | 24 ± 1 | 1 ± 0 | 22 ± 11 | 10 ± 6 |
| FF + MEM | 12 ± 1 | 39 ± 18 | 83 ± 15 | 52 ± 1 | 51 ± 0 | 44 ± 0 | 3 ± 1 | 43 ± 1 | 4 ± 0 | 24 ± 0 | 1 ± 0 | 5 ± 6 | 0 ± 0 |
| FF | 14 ± 1 | 36 ± 11 | 48 ± 1 | 52 ± 1 | 49 ± 0 | 20 ± 0 | 9 ± 5 | 40 ± 0 | 5 ± 0 | 24 ± 0 | 2 ± 0 | -2 ± 1 | 0 ± 0 |

**Holdout-Extrapolate**

| Model | 00. Vis. Goal, Proced. Maze | 01. Transitive Inference | 02. AVM | 03. Cont. Recognition. | 04. Change Detection | 05. Spot Diff Passive | 06. Vis. Goal, Buildings | 07. Spot Diff Basic | 08. Invis. Goal, Buildings | 09. What then Where | 10. Invis. Goal, Empty | 11. Spot Diff Motion | 12. Spot Diff Multi-object |
|---|---|---|---|---|---|---|---|---|---|---|---|---|---|
| MRA: LSTM+MEM+CPC | 9 ± 0 | 122 ± 23 | 99 ± 0 | 98 ± 0 | 92 ± 1 | 78 ± 1 | 15 ± 0 | 77 ± 0 | 8 ± 0 | 31 ± 3 | 1 ± 0 | 45 ± 1 | 10 ± 0 |
| LSTM + MEM + REC | 44 ± 4 | 145 ± 3 | 99 ± 0 | 88 ± 1 | 80 ± 14 | 51 ± 10 | 57 ± 2 | 42 ± 0 | 8 ± 1 | 5 ± 0 | 0 ± 0 | 41 ± 1 | 7 ± 0 |
| LSTM + MEM | 4 ± 1 | 140 ± 7 | 96 ± 1 | 76 ± 13 | 53 ± 1 | 82 ± 4 | 14 ± 0 | 42 ± 0 | 8 ± 1 | 30 ± 2 | 0 ± 0 | 44 ± 0 | 5 ± 3 |
| LSTM + CPC | 2 ± 0 | 83 ± 18 | 47 ± 1 | 50 ± 0 | 91 ± 0 | 74 ± 0 | 11 ± 0 | 47 ± 1 | 6 ± 0 | 26 ± 6 | 0 ± 0 | 43 ± 1 | 9 ± 0 |
| LSTM | 4 ± 1 | 87 ± 21 | 43 ± 2 | 50 ± 0 | 54 ± 0 | 83 ± 0 | 11 ± 0 | 44 ± 1 | 5 ± 0 | 35 ± 1 | 0 ± 0 | 42 ± 1 | 5 ± 2 |
| FF + MEM + REC | 29 ± 14 | 134 ± 49 | 98 ± 1 | 88 ± 1 | 53 ± 0 | 44 ± 1 | 44 ± 5 | 43 ± 0 | 0 ± 0 | 13 ± 0 | 1 ± 0 | 42 ± 0 | 1 ± 1 |
| FF + MEM + CPC | 6 ± 1 | 36 ± 6 | 98 ± 1 | 96 ± 0 | 79 ± 1 | 68 ± 1 | 44 ± 2 | 67 ± 6 | 1 ± 0 | 16 ± 1 | 0 ± 0 | 22 ± 11 | 4 ± 2 |
| FF + MEM | 5 ± 1 | 44 ± 13 | 80 ± 18 | 51 ± 1 | 53 ± 1 | 48 ± 4 | 1 ± 0 | 41 ± 0 | 2 ± 0 | 17 ± 0 | 1 ± 0 | 6 ± 6 | 0 ± 0 |
| FF | 7 ± 1 | 28 ± 9 | 48 ± 1 | 52 ± 0 | 53 ± 1 | 13 ± 12 | 0 ± 1 | 41 ± 1 | 1 ± 0 | 17 ± 0 | 0 ± 0 | -2 ± 0 | 0 ± 0 |

Figure 14: Heatmap of ablations per task including standard errors. Tasks are sorted by normalized score across models during training, such that the task with the highest mean scores in training is in the leftmost column, and the model that had the highest mean scores in training is at the top row.

# B  Human-Normalized Scores and Episode Rewards

We used one action set across all PsychLab tasks, and another across the 3D tasks.

In PsychLab we used a set of five actions: look left, look right, look up, look down, do nothing.

For the rest, we used a set of eight actions: move forward, move backward, strafe left, strafe right, look left, look right, look left while moving forward, look right while moving forward.

In Figure 13 we show the training and test curves for each of our ablation models on all tasks. The curves in bold correspond to the median score across three random seeds, and the corresponding confidence intervals are shown in lighter shades.

## B.1  Human-Normalized Score Computation

We computed the Human-Normalized Scores used in our heatmap via the following procedure. In our reported results we used three seeds, and took a rolling average as described below.

1. For each seed, apply smoothing in the form of exponential weighted moving average[3].
2. For each seed, take a further rolling average of the episode reward, over a window of 10.
3. Among these rolling reward windows, find the highest window value over the course of training. The mean over the seeds corresponds to $R_{Level=Train}$.
4. For each seed, find the time-step that corresponds to $R_{Level=Train}$, to use as a snapshot point for comparison against the holdout levels.
5. At this snapshot point, record the seed-averaged rolling episode reward for the two holdout levels, $R_{Level=Holdout-Interpolate}$ and $R_{Level=Holdout-Extrapolate}$.
6. Obtain the episode reward of a random agent $R_{Random}$ and the episode reward achieved by a human, $R_{Human}$.
7. For Train, Holdout-Interpolate, and Holdout-Extrapolate, with corresponding standard error:

$$HumanNormalizedScore = \frac{R_{Level=\cdot} - R_{Random}}{R_{Human} - R_{Random}} * 100 \qquad (6)$$

Results are shown ranked (best at top) in Figure 3.

Table 3: Ranking of ablation models, sorted by overall task-averaged human-normalized score.

| Model | Average Human-Normalized Score (percentage points) | | |
|---|---|---|---|
| | Train | Holdout-Interpolate | Holdout-Extrapolate |
| MRA: LSTM + MEM + CPC | $92.9 \pm 3.9$ | $56.2 \pm 5.8$ | $52.6 \pm 6.5$ |
| LSTM + MEM + REC | $82.2 \pm 5.2$ | $54.2 \pm 2.3$ | $51.4 \pm 4.9$ |
| LSTM + MEM | $78.7 \pm 5.8$ | $50.0 \pm 3.1$ | $45.8 \pm 4.5$ |
| LSTM + CPC | $77.6 \pm 4.6$ | $42.7 \pm 2.8$ | $37.7 \pm 5.3$ |
| FF + MEM + REC | $63.1 \pm 9.6$ | $45.4 \pm 3.7$ | $45.4 \pm 14.2$ |
| FF + MEM + CPC | $62.6 \pm 5.9$ | $45.4 \pm 7.3$ | $41.3 \pm 4.0$ |
| LSTM | $73.0 \pm 6.9$ | $40.2 \pm 4.3$ | $35.6 \pm 5.9$ |
| FF + MEM | $42.3 \pm 5.8$ | $27.8 \pm 6.7$ | $27.0 \pm 6.6$ |
| FF | $33.9 \pm 3.3$ | $23.0 \pm 3.5$ | $19.7 \pm 4.2$ |

## B.2  Episode Rewards

Absolute episode rewards per task per level, obtained by trained agent as well as $R_{Random}$ and $R_{Human}$, with standard error[4] bars. See Tables 4 to 16.

## C Model

### C.1 Importance Weighted Actor-Learner Architecture

We use the Importance Weighted Actor-Learner Architecture (IMPALA) (Espeholt et al., 2018) in our work. IMPALA uses an off-policy actor-critic approach where decoupled actors communicate experience to a learner. The actor-to-learner relationship is many-to-one. Each actor generates a batched trajectory, or episode, of experience and sends the state-action-reward traces $(s_0, a_0, r_0, ..., s_T, a_T, r_T)$ to its respective learner. The learner gathers trajectories from each actor and computes gradients to update the model parameters continuously. As actors finish processing a trajectory they receive parameter updates from the learner then continue to generate trajectories.

Under this scheme the actors and learner policies fall out of sync between parameter updates. The actor's *behaviour policy*, $\mu$, is said to have *policy lag* with respect to the *target policy* of the learner, $\pi$. To correct for this effect importance weighting with *V-trace* targets are computed for each step:

$$v_s \stackrel{d}{=} V(x_s) + \sum_{t=s}^{s+T-1} \gamma^{t-s} (\pi_{i=s}^{t-1} c_i) \rho_t (r_t + \gamma V(x_{t+1}) - V(x_t)) \tag{7}$$

where $\gamma \in [0, 1)$ is a discount factor, $x_t$ and $r_t$ are the state reward at time-step $t$, $\rho_t = min(\bar{\rho}, \frac{\pi(a_t|x_t)}{\mu(a_t|x_t)})$ and $c_i = min(\bar{c}, \frac{\pi(a_i|x_i)}{\mu(a_i|x_i)})$ are truncated importance sampling weights. These V-trace targets are used to compute gradients for the policy approximation in the learner. This enables observations and parameters to each flow in a single direction, allowing for high data efficiency and resource allocation in comparison to other other approaches, such as asynchronous advantageous actor critic (A3C) (Mnih et al., 2016).

### C.2 Residual Network Architecture

To process the pixel input, the Memory Recall Agent and the other baselines reported in this work use a residual network (He et al., 2015) with a similar architecture found in (Espeholt et al., 2018). This consists of three convolutional blocks with feature map counts of size 16, 32, and 32; each block has a convolutional layer with kernel size 3x3 followed max pooling with kernel size 3x3 and stride 2x2, followed by two residual subblocks. The output from the top residual block is followed by a 256-unit MLP to generate latent representations $x_t$ to be passed to the working memory and query network $f_k$.

### C.3 Contrastive Predictive Coding

We use the encoder already present in the agent's architecture, the convolution neural network that takes the input frame $(i_t)$ and converts it to the embedded visual input $x_t$. The auto-regressive component is the working memory itself, which takes $x_t$ as input and outputs $h_t$ which can be used to predict future steps in latent space: $x_{t+1}, \ldots, x_{t+N}$, where $N$ said to be the number of CPC steps. Figure 4(b) illustrates the CPC approach (van den Oord et al. (2018)).

To introduce a noise-contrastive loss the mutual information (Eq. 8) between the target encoded representations $x_t$, and the contexts $(c_t)$ – which in our case are the memory states $h_t$. For each sample, a positive real score is then generated via $f_k$, a log-bilinear density function (Eq. 9) by taking the current output from the working memory $h_t$ and the latent vector of the $k^{th}$ step, $x_{t+k}$.

$$I(x, c) = \sum_{x,c} p(x, c) \log \frac{p(x|c)}{p(x)} \tag{8}$$

$$f_k(x_{t+k}, h_t) = \exp\left(x_{t+k}^T W_k h_t\right) \tag{9}$$

Given a sample trajectory of length $T$ and a fixed number of maximum CPC steps $N \leq T - 1$, predictions are computed for each of the $k$-step predictive models ($1 \leq k \leq N$). For timestep $t$ ($1 \leq t \leq T - k$) and predictive model $k$ let $\chi_{t,k}$ denote a set of samples from which a contrastive noise estimate is derived. Each set $\chi_{t,k}$ may be split into two subsets, a single positive sample and $T - k - 1$ negative samples: $\chi_{t,k}^+ = \{x_{t+k}\}$, $\chi_{t,k}^- = \{x_{k+1}, ..., x_{t+k-1}, x_{t+k+1}, ..., x_{T-k}\}$ such that $\chi_{t,k} = \chi_{t,k}^+ + \chi_{t,k}^-$ ($|\chi_{t,k}| = T - k$). The noise

contrastive loss is then determined by computing the categorical cross-entropy over the $(t, k)$-trajectory sample set $\forall(t, k)$. Details can be seen in Eq. 10.

$$\mathcal{L}_{CPC(\chi_{t,k})} = \mathbb{E}_{\chi_{t,k}^+} \left[ \log \frac{f_k(x_{t+k}, h_t)}{\sum_{i_j \epsilon \chi_{t,k}} f_k(x_j, h_t)} \right] + \mathbb{E}_{\chi_{t,k}^-} \left[ \log \left( 1 - \frac{f_k(x_{t+k}, h_t)}{\sum_{i_j \epsilon \chi_{t,k}} f_k(x_j, h_t)} \right) \right] \qquad (10)$$

### C.4 Reconstruction

Action and reward reconstructions are linear projections $f_r(r_t) = r_t W_r + b_r$ and $f_a(a_t) = a_t W_a + b_a$ while reconstructions of the image input $i_t$ are generated via the transpose residual network $f_{RN}^T$. Sum of squared error losses are used for prior step reward and prior step actions while sigmoid cross-entropy is used for the image reconstruction. The losses are summed and scaled by a cost hyper-parameter for each to produce a full reconstruction loss for the model, $\mathcal{L}_{\text{REC}}$. See equations 11 to 14 below for more details ($\sigma$ is the sigmoid function).

$$\mathcal{L}_{\text{reward}} = \frac{\sum_{i=1}^{T} (r_{t-1,i} - f(r_{t-1,i}))^2}{2} \qquad (11)$$

$$\mathcal{L}_{\text{action}} = \frac{\sum_{i=1}^{T} (a_{t-1,i} - f(a_{t-1,i}))^2}{2} \qquad (12)$$

$$\mathcal{L}_{\text{image}} = -i_t \log(\sigma(f_{RN}^T(h_t))) - (1 - i_t) \log \left( 1 - \sigma(f_{RN}^T(h_t)) \right) \qquad (13)$$

$$\mathcal{L}_{\text{REC}} = c_{\text{image}} \mathcal{L}_{\text{image}} + c_{\text{action}} \mathcal{L}_{\text{action}} + c_{\text{reward}} \mathcal{L}_{\text{reward}} \qquad (14)$$

In our experiments we set $c_{\text{image}} = c_{\text{action}} = c_{\text{reward}} = 1.0$ for all tasks, except in AVM, Continuous Recognition and Change Detection, where $c_{\text{image}} = 30, 1.5$, and $3$, respectively. We did not tune for this hyper-parameter, we used first guess or previous work (such as (Wayne et al., 2018)) for choosing it.

## D Hyper-parameter Tuning

All experiments used three seeds, with identical hyper-parameters each. Given the scope of the experiments undertaken, all hyper-parameter tuning was preliminary and not exhaustive.

Initial hyper-parameters were either inherited from the IMPALA paper or given an arbitrary first-guess value that seemed reasonable. Whatever tuning that was done was performed in a relatively systematic way: Hyper-parameters were shared across all model variations, and tuned with the objective of getting as many model variations as possible to achieve adequate performance on the training tasks.

The PsychLab tasks were the ones with the most tuning. For PsychLab, we performed a manual sweep over arbitrary reasonable-seeming values when train performance wasn't getting off the floor or was too noisy. We had a preference for hypers that fared well across all models (e.g. choosing a bigger hidden size of 1024 rather than 512 for the controller so that FF models would have capacity).

For the other tasks, very minimal tuning occurred and hyper-parameters were first-guess. With Spot the Difference, we tried two different discount rates and went with the better one. For Goal Navigation and Transitive Inference tasks, we stuck to a standardized discount rate of 0.99.

We did not perform any tuning for REC throughout.

**Fixed hyper-parameters** (See Table 17) For optimizers, whenever we used Adam we standardized the discount rate to 0.98, and whenever we used RMSProp the discount rate was mostly 0.99 except in certain cases where we were able to also try 0.999 and found that it did better. Whenever we used an external episodic memory module ('MEM') we used the fixed hyper-parameters in Table 17.

For individual task hyper-parameter configurations see Table 18.

Table 4: Episode reward: PsychLab - AVM

| Model | Train | Holdout-Interpolate | Holdout-Extrapolate |
|---|---|---|---|
| FF | $25.90 \pm 0.32$ | $19.37 \pm 0.43$ | $35.74 \pm 0.37$ |
| FF + MEM | $43.14 \pm 6.12$ | $33.34 \pm 5.92$ | $60.16 \pm 13.56$ |
| FF + MEM + CPC | $49.98 \pm 0.00$ | $38.86 \pm 0.32$ | $73.66 \pm 0.63$ |
| FF + MEM + REC | $50.00 \pm 0.00$ | $39.76 \pm 0.14$ | $73.17 \pm 0.46$ |
| LSTM | $33.35 \pm 0.24$ | $27.20 \pm 0.25$ | $32.34 \pm 1.20$ |
| LSTM + CPC | $30.75 \pm 0.21$ | $25.64 \pm 0.26$ | $35.50 \pm 1.09$ |
| LSTM + MEM | $50.00 \pm 0.00$ | $39.99 \pm 0.01$ | $72.32 \pm 0.98$ |
| LSTM + MEM + REC | $50.00 \pm 0.00$ | $39.63 \pm 0.36$ | $73.91 \pm 0.28$ |
| MRA: LSTM+MEM+CPC | $50.00 \pm 0.00$ | $39.99 \pm 0.00$ | $74.32 \pm 0.13$ |
| Random | $0.06 \pm 0.00$ | $0.06 \pm 0.00$ | $0.06 \pm 0.00$ |
| Human | $50.00 \pm 0.00$ | $40.00 \pm 0.00$ | $75.00 \pm 0.00$ |

Table 5: Episode reward: PsychLab - Continuous Recognition

| Model | Train | Holdout-Interpolate | Holdout-Extrapolate |
|---|---|---|---|
| FF | $26.90 \pm 0.28$ | $20.62 \pm 0.28$ | $38.45 \pm 0.34$ |
| FF + MEM | $26.51 \pm 0.06$ | $20.54 \pm 0.49$ | $37.96 \pm 0.51$ |
| FF + MEM + CPC | $49.60 \pm 0.01$ | $39.51 \pm 0.15$ | $71.40 \pm 0.15$ |
| FF + MEM + REC | $49.78 \pm 0.08$ | $39.90 \pm 0.03$ | $65.57 \pm 0.56$ |
| LSTM | $27.11 \pm 0.29$ | $20.92 \pm 0.11$ | $37.28 \pm 0.18$ |
| LSTM + CPC | $26.25 \pm 0.26$ | $20.11 \pm 0.55$ | $37.46 \pm 0.36$ |
| LSTM + MEM | $42.18 \pm 6.93$ | $39.68 \pm 0.06$ | $56.59 \pm 9.84$ |
| LSTM + MEM + REC | $49.78 \pm 0.08$ | $39.90 \pm 0.03$ | $65.57 \pm 0.56$ |
| MRA: LSTM+MEM+CPC | $49.92 \pm 0.03$ | $39.83 \pm 0.00$ | $72.52 \pm 0.25$ |
| Random | $0.04 \pm 0.00$ | $0.05 \pm 0.00$ | $0.05 \pm 0.00$ |
| Human | $49.40 \pm 0.24$ | $39.40 \pm 0.40$ | $74.20 \pm 0.58$ |

Table 6: Episode reward: PsychLab - Change Detection

| Model | Train | Holdout-Interpolate | Holdout-Extrapolate |
|---|---|---|---|
| FF | $26.40 \pm 0.08$ | $24.17 \pm 0.10$ | $24.73 \pm 0.48$ |
| FF + MEM | $25.76 \pm 0.16$ | $24.95 \pm 0.12$ | $24.97 \pm 0.36$ |
| FF + MEM + CPC | $44.76 \pm 0.05$ | $36.07 \pm 0.46$ | $36.95 \pm 0.40$ |
| FF + MEM + REC | $25.82 \pm 0.22$ | $23.99 \pm 0.29$ | $24.89 \pm 0.23$ |
| LSTM | $26.39 \pm 0.21$ | $25.24 \pm 0.43$ | $25.37 \pm 0.22$ |
| LSTM + CPC | $48.37 \pm 0.39$ | $41.43 \pm 0.12$ | $42.72 \pm 0.11$ |
| LSTM + MEM | $26.21 \pm 0.10$ | $24.77 \pm 0.28$ | $24.63 \pm 0.52$ |
| LSTM + MEM + REC | $39.12 \pm 5.88$ | $42.12 \pm 1.97$ | $37.31 \pm 6.38$ |
| MRA: LSTM+MEM+CPC | $49.14 \pm 0.24$ | $42.24 \pm 0.07$ | $43.00 \pm 0.39$ |
| Random | $0.00 \pm 0.00$ | $0.00 \pm 0.00$ | $0.00 \pm 0.00$ |
| Human | $47.60 \pm 0.40$ | $48.80 \pm 0.58$ | $46.80 \pm 1.07$ |

Table 7: Episode reward: PsychLab - What Then Where

| Model | Train | Holdout-Interpolate | Holdout-Extrapolate |
|---|---|---|---|
| FF | $12.71 \pm 0.06$ | $12.19 \pm 0.06$ | $8.39 \pm 0.16$ |
| FF + MEM | $12.11 \pm 0.14$ | $12.05 \pm 0.12$ | $8.34 \pm 0.11$ |
| FF + MEM + CPC | $12.92 \pm 0.32$ | $12.21 \pm 0.20$ | $7.73 \pm 0.29$ |
| FF + MEM + REC | $6.54 \pm 0.50$ | $6.10 \pm 0.35$ | $6.30 \pm 0.10$ |
| LSTM | $37.18 \pm 0.14$ | $25.06 \pm 0.34$ | $17.51 \pm 0.38$ |
| LSTM + CPC | $24.21 \pm 6.04$ | $20.68 \pm 3.89$ | $12.99 \pm 2.79$ |
| LSTM + MEM | $26.19 \pm 6.23$ | $23.74 \pm 2.65$ | $14.72 \pm 1.22$ |
| LSTM + MEM + REC | $2.96 \pm 0.04$ | $1.71 \pm 0.27$ | $2.34 \pm 0.22$ |
| MRA: LSTM+MEM+CPC | $24.22 \pm 5.45$ | $23.10 \pm 1.82$ | $15.54 \pm 1.39$ |
| Random | $0.02 \pm 0.00$ | $0.03 \pm 0.00$ | $0.01 \pm 0.00$ |
| Human | $50.00 \pm 0.00$ | $50.00 \pm 0.00$ | $49.60 \pm 0.24$ |

Table 8: Episode reward: Spot Diff Basic

| Model | Train | Holdout-Interpolate | Holdout-Extrapolate |
|---|---|---|---|
| FF | $0.46 \pm 0.00$ | $0.43 \pm 0.00$ | $0.43 \pm 0.01$ |
| FF + MEM | $0.46 \pm 0.01$ | $0.45 \pm 0.01$ | $0.44 \pm 0.00$ |
| FF + MEM + CPC | $0.93 \pm 0.02$ | $0.71 \pm 0.04$ | $0.69 \pm 0.05$ |
| FF + MEM + REC | $0.44 \pm 0.01$ | $0.45 \pm 0.02$ | $0.45 \pm 0.00$ |
| LSTM | $0.46 \pm 0.01$ | $0.45 \pm 0.01$ | $0.46 \pm 0.01$ |
| LSTM + CPC | $0.54 \pm 0.03$ | $0.48 \pm 0.03$ | $0.49 \pm 0.01$ |
| LSTM + MEM | $0.47 \pm 0.00$ | $0.45 \pm 0.01$ | $0.45 \pm 0.00$ |
| LSTM + MEM + REC | $0.46 \pm 0.01$ | $0.44 \pm 0.01$ | $0.45 \pm 0.00$ |
| MRA: LSTM+MEM+CPC | $0.90 \pm 0.07$ | $0.81 \pm 0.00$ | $0.78 \pm 0.00$ |
| Random | $0.05 \pm 0.00$ | $0.04 \pm 0.00$ | $0.04 \pm 0.00$ |
| Human | $1.00 \pm 0.00$ | $1.00 \pm 0.00$ | $1.00 \pm 0.00$ |

Table 9: Episode reward: Spot Diff Passive

| Model | Train | Holdout-Interpolate | Holdout-Extrapolate |
|---|---|---|---|
| FF | $0.22 \pm 0.09$ | $0.23 \pm 0.00$ | $0.14 \pm 0.11$ |
| FF + MEM | $0.54 \pm 0.05$ | $0.46 \pm 0.00$ | $0.49 \pm 0.04$ |
| FF + MEM + CPC | $0.80 \pm 0.01$ | $0.66 \pm 0.03$ | $0.68 \pm 0.01$ |
| FF + MEM + REC | $0.45 \pm 0.00$ | $0.45 \pm 0.00$ | $0.45 \pm 0.01$ |
| LSTM | $0.95 \pm 0.00$ | $0.85 \pm 0.01$ | $0.84 \pm 0.00$ |
| LSTM + CPC | $0.91 \pm 0.01$ | $0.77 \pm 0.00$ | $0.75 \pm 0.00$ |
| LSTM + MEM | $0.97 \pm 0.01$ | $0.78 \pm 0.04$ | $0.83 \pm 0.03$ |
| LSTM + MEM + REC | $0.74 \pm 0.12$ | $0.54 \pm 0.06$ | $0.52 \pm 0.09$ |
| MRA: LSTM+MEM+CPC | $0.96 \pm 0.01$ | $0.82 \pm 0.01$ | $0.78 \pm 0.01$ |
| Random | $0.03 \pm 0.00$ | $0.03 \pm 0.00$ | $0.02 \pm 0.00$ |
| Human | $1.00 \pm 0.00$ | $1.00 \pm 0.00$ | $1.00 \pm 0.00$ |

Table 10: Episode reward: Spot Diff Multi-Object

| Model | Train | Holdout-Interpolate | Holdout-Extrapolate |
|---|---|---|---|
| FF | $0.02 \pm 0.01$ | $0.01 \pm 0.00$ | $0.00 \pm 0.00$ |
| FF + MEM | $0.01 \pm 0.00$ | $0.01 \pm 0.00$ | $0.00 \pm 0.00$ |
| FF + MEM + CPC | $0.12 \pm 0.08$ | $0.11 \pm 0.06$ | $0.04 \pm 0.02$ |
| FF + MEM + REC | $0.18 \pm 0.02$ | $0.17 \pm 0.01$ | $0.01 \pm 0.01$ |
| LSTM | $0.52 \pm 0.20$ | $0.14 \pm 0.07$ | $0.05 \pm 0.02$ |
| LSTM + CPC | $0.58 \pm 0.03$ | $0.24 \pm 0.01$ | $0.09 \pm 0.00$ |
| LSTM + MEM | $0.39 \pm 0.05$ | $0.14 \pm 0.07$ | $0.05 \pm 0.03$ |
| LSTM + MEM + REC | $0.18 \pm 0.04$ | $0.15 \pm 0.02$ | $0.07 \pm 0.00$ |
| MRA: LSTM+MEM+CPC | $0.69 \pm 0.02$ | $0.27 \pm 0.00$ | $0.10 \pm 0.00$ |
| Random | $0.01 \pm 0.00$ | $0.01 \pm 0.00$ | $0.00 \pm 0.00$ |
| Human | $1.00 \pm 0.00$ | $1.00 \pm 0.00$ | $1.00 \pm 0.00$ |

Table 11: Episode reward: Spot Diff Motion

| Model | Train | Holdout-Interpolate | Holdout-Extrapolate |
|---|---|---|---|
| FF | $0.12 \pm 0.04$ | $0.00 \pm 0.01$ | $0.00 \pm 0.00$ |
| FF + MEM | $0.08 \pm 0.05$ | $0.07 \pm 0.05$ | $0.08 \pm 0.06$ |
| FF + MEM + CPC | $0.13 \pm 0.09$ | $0.24 \pm 0.10$ | $0.23 \pm 0.11$ |
| FF + MEM + REC | $0.46 \pm 0.01$ | $0.42 \pm 0.01$ | $0.43 \pm 0.00$ |
| LSTM | $0.45 \pm 0.00$ | $0.44 \pm 0.01$ | $0.43 \pm 0.01$ |
| LSTM + CPC | $0.46 \pm 0.00$ | $0.43 \pm 0.00$ | $0.44 \pm 0.01$ |
| LSTM + MEM | $0.45 \pm 0.01$ | $0.46 \pm 0.00$ | $0.45 \pm 0.00$ |
| LSTM + MEM + REC | $0.44 \pm 0.02$ | $0.44 \pm 0.01$ | $0.42 \pm 0.01$ |
| MRA: LSTM+MEM+CPC | $0.47 \pm 0.01$ | $0.45 \pm 0.00$ | $0.46 \pm 0.01$ |
| Random | $0.02 \pm 0.00$ | $0.02 \pm 0.00$ | $0.02 \pm 0.00$ |
| Human | $1.00 \pm 0.00$ | $1.00 \pm 0.00$ | $1.00 \pm 0.00$ |

Table 12: Episode reward: Visible Goal With Buildings

| Model | Train | Holdout-Interpolate | Holdout-Extrapolate |
|---|---|---|---|
| FF | 12.27 ± 0.83 | 3.74 ± 1.70 | 0.14 ± 0.14 |
| FF + MEM | 13.58 ± 1.45 | 1.52 ± 0.28 | 0.47 ± 0.02 |
| FF + MEM + CPC | 31.87 ± 0.25 | 22.99 ± 0.70 | 9.83 ± 0.40 |
| FF + MEM + REC | 31.01 ± 0.77 | 23.42 ± 0.33 | 9.84 ± 1.09 |
| LSTM | 28.72 ± 0.34 | 11.37 ± 0.00 | 2.66 ± 0.00 |
| LSTM + CPC | 29.46 ± 0.23 | 11.52 ± 0.08 | 2.52 ± 0.02 |
| LSTM + MEM | 30.92 ± 0.28 | 11.74 ± 0.22 | 3.29 ± 0.10 |
| LSTM + MEM + REC | 35.95 ± 0.28 | 25.16 ± 0.05 | 12.54 ± 0.34 |
| MRA: LSTM+MEM+CPC | 32.45 ± 0.39 | 12.66 ± 0.06 | 3.38 ± 0.05 |
| Random | 1.08 ± 0.02 | 0.58 ± 0.01 | 0.22 ± 0.01 |
| Human | 23.60 ± 1.69 | 23.50 ± 0.75 | 14.30 ± 0.60 |

Table 13: Episode reward: Invisible Goal With Buildings

| Model | Train | Holdout-Interpolate | Holdout-Extrapolate |
|---|---|---|---|
| FF | 9.30 ± 0.33 | 1.80 ± 0.01 | 0.28 ± 0.02 |
| FF + MEM | 9.95 ± 0.65 | 1.54 ± 0.02 | 0.48 ± 0.00 |
| FF + MEM + CPC | 10.65 ± 0.28 | 1.52 ± 0.10 | 0.31 ± 0.01 |
| FF + MEM + REC | 12.29 ± 0.24 | 0.70 ± 0.05 | 0.20 ± 0.00 |
| LSTM | 27.22 ± 0.48 | 2.01 ± 0.19 | 0.79 ± 0.01 |
| LSTM + CPC | 28.46 ± 0.63 | 2.25 ± 0.01 | 0.86 ± 0.01 |
| LSTM + MEM | 30.15 ± 0.04 | 3.10 ± 0.08 | 1.17 ± 0.08 |
| LSTM + MEM + REC | 32.10 ± 0.05 | 2.75 ± 0.24 | 1.17 ± 0.09 |
| MRA: LSTM+MEM+CPC | 30.51 ± 0.21 | 2.39 ± 0.09 | 1.09 ± 0.03 |
| Random | 0.95 ± 0.02 | 0.53 ± 0.01 | 0.20 ± 0.01 |
| Human | 17.37 ± 1.91 | 12.40 ± 1.45 | 4.90 ± 0.71 |

Table 14: Episode reward: Invisible Goal Empty Arena

| Model | Train | Holdout-Interpolate | Holdout-Extrapolate |
|---|---|---|---|
| FF | 1.78 ± 0.11 | 0.38 ± 0.04 | 0.05 ± 0.00 |
| FF + MEM | 2.21 ± 0.05 | 0.28 ± 0.02 | 0.07 ± 0.01 |
| FF + MEM + CPC | 2.37 ± 0.07 | 0.22 ± 0.02 | 0.05 ± 0.00 |
| FF + MEM + REC | 3.25 ± 0.25 | 0.27 ± 0.02 | 0.06 ± 0.01 |
| LSTM | 7.60 ± 0.14 | 0.14 ± 0.01 | 0.05 ± 0.01 |
| LSTM + CPC | 10.48 ± 0.35 | 0.12 ± 0.02 | 0.03 ± 0.01 |
| LSTM + MEM | 10.32 ± 0.12 | 0.19 ± 0.02 | 0.03 ± 0.01 |
| LSTM + MEM + REC | 12.40 ± 0.08 | 0.08 ± 0.01 | 0.04 ± 0.00 |
| MRA: LSTM+MEM+CPC | 13.04 ± 0.60 | 0.23 ± 0.04 | 0.07 ± 0.01 |
| Random | 0.15 ± 0.01 | 0.15 ± 0.01 | 0.03 ± 0.00 |
| Human | 4.90 ± 1.32 | 1.70 ± 0.67 | 0.30 ± 0.30 |

Table 15: Episode reward: Visible Goal Procedural Maze

| Model | Train | Holdout-Interpolate | Holdout-Extrapolate |
|---|---|---|---|
| FF | 174.63 ± 4.27 | 43.55 ± 3.08 | 11.93 ± 2.17 |
| FF + MEM | 224.53 ± 11.31 | 37.80 ± 2.28 | 8.52 ± 0.87 |
| FF + MEM + CPC | 272.99 ± 5.31 | 33.38 ± 1.51 | 9.99 ± 1.10 |
| FF + MEM + REC | 607.48 ± 36.98 | 59.64 ± 9.41 | 43.07 ± 19.35 |
| LSTM | 463.43 ± 12.84 | 27.09 ± 1.69 | 6.52 ± 1.86 |
| LSTM + CPC | 473.42 ± 3.84 | 19.72 ± 0.98 | 4.90 ± 0.36 |
| LSTM + MEM | 523.14 ± 10.52 | 22.39 ± 3.65 | 7.83 ± 1.25 |
| LSTM + MEM + REC | 655.10 ± 11.65 | 49.53 ± 24.78 | 57.21 ± 23.92 |
| MRA: LSTM+MEM+CPC | 546.08 ± 2.26 | 40.64 ± 0.00 | 14.21 ± 0.00 |
| Random | Small: 7.79 ± 0.14 Large: 1.97 ± 0.06 | 3.89 ± 0.10 | 1.19 ± 0.05 |
| Human | Small: 364.00 ± 43.20 Large: 104.00 ± 23.58 | 198.00 ± 24.98 | 86.00 ± 20.15 |

Table 16: Episode reward: Transitive Inference

| Model | Train | Holdout-Interpolate | Holdout-Extrapolate |
|---|---|---|---|
| FF | 3.71 ± 0.07 | 3.52 ± 0.06 | 3.73 ± 0.04 |
| FF + MEM | 4.34 ± 0.51 | 4.09 ± 0.48 | 5.76 ± 0.18 |
| FF + MEM + CPC | 4.64 ± 0.76 | 5.16 ± 0.01 | 5.62 ± 0.41 |
| FF + MEM + REC | 0.47 ± 0.22 | 0.30 ± 0.01 | 0.69 ± 0.12 |
| LSTM | 8.67 ± 0.55 | 5.31 ± 0.10 | 7.32 ± 0.38 |
| LSTM + CPC | 9.65 ± 0.53 | 5.59 ± 0.02 | 7.77 ± 0.08 |
| LSTM + MEM | 8.98 ± 0.42 | 6.76 ± 0.75 | 8.84 ± 0.41 |
| LSTM + MEM + REC | 10.86 ± 0.02 | 8.88 ± 0.11 | 9.80 ± 0.16 |
| MRA: LSTM+MEM+CPC | 10.34 ± 0.10 | 7.21 ± 0.19 | 9.81 ± 0.09 |
| Random | Small: 1.44 ± 0.02 Large: 1.44 ± 0.02 | 1.42 ± 0.02 | 1.43 ± 0.02 |
| Human | Small: 5.40 ± 2.20 Large: 6.60 ± 2.69 | 6.00 ± 2.45 | 7.20 ± 2.94 |

Table 17: Fixed hyper-parameters

| Optimizer | |
|---|---|
| Adam: | |
| Beta1 | 0.9 |
| Beta2 | 0.999 |
| Epsilon | 1e-4 |
| RMSProp: | |
| Epsilon | 0.1 |
| Momentum (Inherited from IMPALA paper) | 0.0 |
| Decay | 0.99 |
| **MEM** | |
| Number of k-nearest neighbors to retrieve from MEM | 10 |
| MEM key size (and accordingly, query size) | 128 |
| Capacity (max number of timesteps storable) | 2048 for Unity levels, else 1024 |

Table 18: Hyper-parameters

| Parameter | Hidden size | Baseline cost[5] | Entropy | Batch size | Unroll length | Discount | Optimizer | Learning rate | Num CPC steps | CPC weight |
|---|---|---|---|---|---|---|---|---|---|---|
| **PsychLab** | | | | | | | | | | |
| AVM | 512 | 0.5 | 0.00520[6] | 16 | 50 | 0.98 | Adam | 1e-5 | 10 | 10 |
| Cont. Recognition | 1024 | 0.5 | 0.01 | 16 | 50 | 0.98 | Adam | 1e-5 | 10 | 10 |
| Change Detection | 512 | 0.5 | 0.01 | 16 | 50 | 0.98 | Adam | 1e-5 | 10 | 10 |
| What Then Where | 1024 | 2.0 | 0.01 | 32 | 100 | 0.98 | Adam | 1e-5 | 10 | 30 |
| | | Sweep [0.5, 1.0, 2.0] | | | | | | | | Sweep [10, 30] |
| | | | | | | | | | | |
| **Spot Diff** | | | | | | [7] | | | | |
| Basic | 1024 | 0.5 | 0.003 | 16 | 200 | 0.99 | RMSProp | 1e-4 | 50 | 20 |
| Passive | 1024 | 0.5 | 0.003 | 16 | 200 | 0.999 | RMSProp | 1e-4 | 50 | 20 |
| Multi-object | 1024 | 0.5 | 0.003 | 16 | 200 | 0.99 | RMSProp | 1e-4 | 50 | 20 |
| Motion | 1024 | 0.5 | 0.003 | 16 | 200 | 0.99[8] | RMSProp | 1e-4 | 50 | 20 |
| | | | | | | | | | | |
| **Goal Navigation** | | | | | | | | | | |
| Visible Goal, Proced. Maze | 512 | 0.5 | 0.00520[9] | 16 | 50 | 0.98 | Adam | 1e-5 | 10 | 5 |
| Visible Goal, With Buildings | 1024 | 0.5 | 0.003 | 16 | 200 | 0.99 | RMSProp | 1e-4 | 50 | 20 |
| Invisible Goal With Buildings | 1024 | 0.5 | 0.003 | 16 | 200 | 0.99 | RMSProp | 1e-4 | 50 | 20 |
| Invisible Goal Empty Arena | 1024 | 0.5 | 0.003 | 16 | 200 | 0.99 | RMSProp | 1e-4 | 50 | 20 |
| **Transitive Inference** | 1024 | 0.5 | 0.003 | 16 | 200 | 0.98 | Adam | 1e-4 | 50 | 20 |

[5] Inherited from IMPALA paper, except for *What Then Where*.
[6] Copied from previous work, not tuned for this paper. 0.01 was slower and noisier.
[7] Sweep over [0.99, .999] throughout.
[8] Sweep over [0.98, .99] throughout.
[9] Copied from AVM

## Footnotes

[3]For PsychLab tasks and *Visible Goal Procedural Maze*, alpha = 0.05. For the rest, alpha = 0.001.

[4]Computed over three seeds for trained agent and for random agent. For human scores, all levels had five trials each except the following: 10 for *Visible Goal with Buildings* and *Invisible Goal Empty Arena*, 19 for the Train level of *Invisible Goal with Buildings* and 20 for the other two levels. The difference was due to time constraints.