[Reviews · NeurIPS 2019]

Reviewer 1



tl;dr: This is a good paper. I recommend acceptance. The authors do a good job of motivating their work, and they contribute a nice experimental section with good results. The ablation study was thorough. Well done! --- Many tasks that might be given to an RL agent are impossible without working memory. This paper presents a suite of tasks which require use of that memory in order to succeed. These tasks are compiled from a variety of other sources, either directly or re-implemented for this suite. They're good tasks. This paper also presents a neural architecture for using both working memory and episodic memory. The working memory is implemented with an LSTM, not unlike IMPALA. The episodic memory, however, writes memory which is indexed into a many dimensional vector space. The paper claims that this type of memory lasts longer than the LSTM memory. The authors make a point of saying that none of the models, including the one presented in the paper, are able to do well on some of the tasks. They also show that none of the models perform well on extrapolated tasks (where the difficulty was increased after train time). I think they're doing this to show that their suite of tasks are challenging and worth trying to learn. There appears to be a marked improvement between agents without episodic memory and agents with episodic memory on the heldout test sets. Also, there is the same improvement between feed forward and LSTM agents (working memory). They did develop a novel architecture, though none of the pieces are particularly novel. However, their ablation tests successfully show that the agents with working memory and episodic memory perform better than similar agents without episodic memory or working memory at both training time and test time. Pros: - Generally easy to read - The neural architecture seems sufficiently novel - Need for both working and episodic memory seems well justified. - Thorough ablation tests Cons: - The formatting for Section 2 is *lousy*. Because figures, figure text, and main text are all over the page, it's hard to keep track of what refers to what. The intro to Section 2 says there are 13 tasks, but it's difficult to keep tally throughout the section. It would be especially helpful if the order of the figures matched the order that the tasks are presented in the main text. I think the direction is good, the experiments is good, and the overall quality is good. I wish they had another diagram which really showed their claims about generalization. For example, rather than showing all the data for the individual tasks in one, it could be nicer to show a graph which combined the information across tasks, or some handpicked results that demonstrated successes and failures (in addition to the data they have given). I didn't feel like their results had much to do with generalization as much as it had to do with the need of memory for different types of tasks. Personally, I would have liked more discussion on the need for different types of memory and how their results backed up the theory/intuition.

Reviewer 2



# Originality The problem of incorporating memory in model-free RL is not new, however there is a general lack of qualitative analysis on the problem due to the lack of clear testbeds (since most current ones might have many confounding elements, or different focus) and baselines. This paper attempts at providing both, and thus makes for a good and original contribution to the NeurIPS community. I also appreciated the focus on testing for generalisation across instances of the tasks, since that is an important metric that is often lacking in published papers in the area. # Quality The work presented is overall of high quality. The technical contribution is theoretically sound, as it is a relatively straightforward combination of existing methods. A satisfactory ablation study was provided, and the method was compared against a state of the art distributed RL algorithm, IMPALA. The authors are mostly careful about their state claims about performance of their method, and they managed to mostly convince me of the quality of the presented testbeds. # Clarity The paper is well written, albeit at times a bit too reliant on the presence of supplementary materials. As this is a common (and not easily addressable) problem with work presenting testbed-baselines pairs, this didn't affect the score too heavily, however the exposition would have gained from strongly focusing on any of the two main contributions. # Significance The problem of incorporating and utilising memory in model-free agents is a relatively strong focus of the RL community, and this work sets out to provide both testbeds and baselines to work towards tackling this important issue. The paper provides some insights on the usefulness of auxiliary reconstruction losses, which confirm and strengthen previous findings. Provided the code and the tasks are successfully released, this paper will make for an important baseline towards the quest to solve this general problem.

Reviewer 3



EDIT: changed my overall score from 6 to 7 in light of author's feedback. Positive/negative things +/-: + clearly written + not all tasks are solved - "We plan to release the full task suite within six months of publication." weakens the article as one of its main contribution is this task suite. Overall a good submission, but I feel like the contribution of the task suite is bigger than the modeling contribution. The delayed release of the task suite a big drawback. Nitpicks: It is weird to describe IMPALA (Importance Weighted Actor-Learner Architecture) an agent: "it would be almost identical to IMPALA" -> "it would be almost identical to the model in Espeholt et al. 2018." (page 4). I applaud trying to make it better with heatmap coloring, but Figure 5 is still a bit hard to read (I don't mean the font size).

[Author Response · NeurIPS 2019]

**Paper: Generalization of Reinforcement Learners with Working and Episodic Memory**

We thank the reviewers for their thoughtful and constructive feedback on our manuscript. We are excited about this
work and glad for their help in improving it.

We can confirm that the full task suite will be released at the time of publication (Reviewers 4 and 5) and will include
videos for each task. This should help both contextualize each task's difficulty and illustrate what it involves.

Reviewer 3 noted the Section 2 task descriptions could be better presented. We have reformatted it so that "the order
of the figures matches the order that the tasks are presented in the main text" as suggested. We moved Figure 1, an
overview of the 3D task types, to the beginning of the section to reduce confusion. Due to space constraints we are
not able to show figures for all 13 tasks in this section, but these per-task figures will be in the Appendix and will be
referenced when describing each task. We also changed our description of IMPALA to match Reviewer 5's suggestion.

Regarding the task suite, Reviewer 4 raised a thoughtful consideration on whether "most of the findings translate when
some confounding elements of the tasks are removed e.g. by making the environment 2D, or by removing the need to
navigate around". Some 3D tasks in the suite already have '2D-like' semi-counterparts that do not require navigation,
which we hope may be able to shed light on whether our findings would hold up if translated from 3D to 2D versions.
Namely, for the *Spot the Difference* tasks (Basic and Passive), removing the 3D partial-observability and navigation
features would produce something close to the PsychLab *Change Detection* task (included in our task suite) in terms
of which aspects of memory and of the agent's other abilities are being evaluated. We consider the PsychLab tasks
'2D-like' because everything is fully observable and the agent has a first-person point of view from a fixed point, without
any need to navigate. Based on our heatmap results, we found that *Spot the Difference: Passive*, which is the simplest
*Spot the Difference* level, was overall harder than *Change Detection* for our ablation models. That said, a full analysis
between these tasks and newly created fully 2D analogs runs the risk of overwhelming a single paper. We thus leave a
full enquiry into this issue for future work.

Reviewer 4 noted they "would have liked to see included in the testbeds versions of environments (or some evaluation
metrics) for which the proposed method fails". Two examples where our agent and baselines perform poorly (as can be
seen in Figure 5) are *Spot the Difference: Motion* and *What Then Where*. We will highlight this more explicitly in our
main text.

On our generalization results, we acknowledge that the heatmap numbers are not as easy to read as we would like
them to be (Reviewer 5). Due to space challenges we also had to relegate our generalization-related plots per task to
the Appendix. We thank Reviewer 3 for their excellent suggestion to make a figure that combines information across
tasks, which we hope can tie in with our findings about generalization more clearly. We have created the attached plot
showing each model's normalized performance on training and holdout, averaged across all tasks, and will include it in
the main text.

Reviewer 5 asked for clarification on 'the differences in
performance between CPC and REC'. Why CPC (con-
trastive predictive coding) was more helpful for the harder
tasks in our suite and REC (reconstruction loss) on the
simpler ones certainly bears further investigation. From
the literature, Guo et al (2018) find that in partially observ-
able environments such as first-person-view navigation
tasks, where each observation provides only a partial and
possibly noisy view of the environment, it is vital for the
agent to learn a representation that encodes its uncertainty
about the underlying state of the environment, and further
find that CPC was more useful in getting the learned rep-
resentation to encode the agent's position and orientation
on visually complex 3D tasks, whereas one-step frame
prediction was more useful on visually simple tasks.

Review 4 asked for some clarification on 'the hypothesis
presented in lines 268-270'. The hypothesis was that CPC
captures subtler differences than REC. From the loss functions of these two methods, it should be clear that while REC
requires features capable of reconstructing the full scene on a per-pixel basis, CPC is satisfied with a representation that
is distinguishable from the alternatives. This is not always a good thing, as this means CPC (and mutual information
maximizers in general) can have a problem with representing high amount of information (see Ozair et al, 2019). But
since only a few bits are needed for the episodic memories in our tasks (e.g. textures and shadows are irrelevant), it's
probable that CPC's representational strategy is superior on the more challenging tasks in our suite.

[Meta-Review · NeurIPS 2019]

The reviewers consider the task suite, the memory-augmented model, and the evaluations to be solid contributions. Please be sure to release the task suite...